# Considering the Role of Behavioural Fatigue in the COVID-19 Lockdown Debates in Great Britain: A Retrospective Analysis of a National Survey Assessing the Relationships between Capability, Opportunity, Motivation, and Behaviour

**DOI:** 10.3390/bs14090852

**Published:** 2024-09-23

**Authors:** Saval Khanal, Kelly Ann Schmidtke, Kaloyan Haralampiev, Ivo Vlaev

**Affiliations:** 1Health Economics Consulting, University of East Anglia, East Anglia, Norwich NR4 7TJ, UK; s.khanal@uea.ac.uk; 2Liberal Arts, College of Arts and Sciences, University of Health Sciences and Pharmacy, St. Louis, MO 63110, USA; 3Department of Sociology, Faculty of Philosophy, Sofia University “St. Kliment Ohridski”, 1504 Sofia, Bulgaria; k_haralampiev@phls.uni-sofia.bg; 4Warwick Business School, University of Warwick, Coventry CV4 7AL, UK

**Keywords:** behaviour, COVID-19, nudge systems thinking, retrospective, United Kingdom

## Abstract

During the COVID-19 pandemic, the term “behavioural fatigue” became the centre of policy debates in Great Britain. These debates involved deciding when to go into lockdown and whether behavioural interventions could be effective. Behavioural interventions can only succeed where people’s Capabilities, Opportunities, and Motivations to perform target behaviours are supported. Our retrospective data analyses examine the relationships between people’s Capabilities, Opportunities, Motivations, and Behaviours, i.e., adherence to lockdown guidelines. Our cross-sectional analyses include 17,962 unique participants in Great Britain who completed a survey over the initial 30 days of the first lockdown (April 2020). We examine trends in responses to each scale and then the relationships between the scales using Granger’s causality test with tests for stationarity and cointegration. A mixture of increasing and decreasing trends was identified for Capabilities and Opportunities. Decreasing trends were identified for Motivation and Behaviour. Granger’s causality tests found that Capability forecasts Opportunity and Behaviour and that Motivation forecasts Opportunity. The discussion reiterates that to realise and maintain Behaviour changes, policies surrounding people’s Capabilities, Opportunities, and Motivations must continue to support target behaviours.

## 1. Background

In the absence of medical interventions, government guidance around the world aimed to decrease COVID-19 infections by changing people’s behaviours [1,2]. How policymakers conceptualise behaviour change inevitably influences the policies they develop. Broadly speaking, two policy strategies could inform government interventions, which, in line with Rose’s seminal work, we refer to as “high-risk prevention strategies” and “population prevention strategies” [3]. According to Rose, high-risk prevention strategies seek to reduce the health impact of the diseases in target areas while allowing population immunity to increase eventually resulting in lower case numbers. Given a high-risk prevention strategy, governments could ask people at higher risk to stay home while those at lower risk could continue traveling to work. In contrast, population strategies seek to reduce the number of infections until transmissions cease naturally or acceptable medical interventions become available. Given a population strategy, governments could encourage everyone to stay home in a national lockdown until an effective vaccine is available.

The present text does not assume that governments explicitly considered Rose’s distinction between high-risk and population strategies before developing policies; rather, we propose that policies could be described accordingly. For instance, governments in Brazil initially applied more high-risk strategies with local lockdowns, and in South Korea initially applied more population strategies with national lockdowns [4,5]. Initially, the government in Great Britain appeared to consider a high-risk strategy, suggesting a more local lockdown approach may be taken when and where breakout infections occurred to quickly contain local outbreaks. On the 9 March 2020, Chief Medical Officer Chris Whitty explained that “Anything we do, we have got to be able to sustain. Once we have started these things, we have to continue them through the peak, and there is a risk that, if we go too early, people will understandably get fatigued, and it will be difficult to sustain this over time” [6]. His choice of the word “fatigue” possibly describes the way governments conceptualised behaviour change, which if true could render rapid local lockdowns a rational policy choice. Specifically, if any behaviour change is likely to stop over time, then policy actions to change behaviour must be carefully timed to coincide with the periods of highest risk.

The word choice of “fatigue” was quickly criticised. On the 16th of March, 681 behavioural scientists penned an open letter arguing against the idea that “behavioural fatigue” had scientific underpinnings and urged the government to adopt actions better aligned with a population strategy [7]. Additionally, Ferguson’s (2020) [8] mathematical models were published comparing the potential health impacts of each strategy. These models suggest that a population strategy could dramatically reduce hospitalisations and deaths. On the 23rd of March, Prime Minister Boris Johnson announced the first national lockdown in Great Britain, wherein people were ordered to stay at home except for the following four purposes [9]: (1) “shopping for basic necessities, as infrequently as possible,” (2) “one form of exercise a day—for example a run, walk, or cycle—alone or with members of your household,” (3) “any medical need, to provide care or to help a vulnerable person,” and (4) “travelling to and from work, but only where this is absolutely necessary and cannot be done from home.”

The open letter penned by behavioural scientists did not say that behavioural fatigue did not exist; rather, they were “not convinced that enough is known about behavioural fatigue or to what extent these insights apply to the current exceptional circumstances.” A more critical review states that behavioural fatigue “is not a real phenomenon: it must be either a naïve construct or a policy contrivance” [10]. A related critique outlines three senses of fatigue (tiredness, impairment, and distress), none of which were supported across the COVID-19 lockdowns [11]. The concept of behavioural fatigue remains contentious, partly due to its vague definition during the early pandemic. Instead of viewing it as an immutable psychological determinant, this paper suggests that behavioural fatigue could be understood as a dynamic outcome shaped by the Capability, Opportunity, and Motivation of individuals—a framework captured by the COM-B model. The COM-B model, developed by Michie et al. [12], posits that Capability (skills and knowledge), Opportunity (external factors), and Motivation (internal processes) interact to influence Behaviour. By applying this model, we can better assess the factors that drive or hinder adherence to public health guidance.

The COM-B model describes these three interrelated factors that must be present at a sufficient level for a target behaviour to occur [13]. Compared to other models of behaviour (e.g., the common-sense models, the health beliefs model [14], or the theory of planned behaviour) [15], the COM-B model offers a more comprehensive assessment [16]. Capability has to do with an individual’s physical (e.g., skills) or psychological (e.g., awareness) capacity to perform the behaviour. Opportunity has to do with the physical environment (e.g., objects) and social (e.g., social norms) factors that lie outside the individuals that may enable or prompt them to perform the behaviour. Motivation has to do with an individual’s reflective mental processes (e.g., planning) and automatic mental processes (e.g., emotions) that energise the behaviour.

The 2011 COM-B model is redrawn from Michie’s paper in Figure 1. The model anticipates that all three components influence behaviour. It further specifies that Capability and Opportunity influence Motivation (but not the reverse). And lastly, feedback loops are anticipated, where the performance (or non-performance) of a target Behaviour may influence ongoing Capabilities, Opportunities, and Motivations. The model is largely used to understand individual behaviour. The present paper extends the model to population trends.

The present paper assesses data collected about people’s Capabilities, Opportunities, and Motivations to adhere to government guidance (Behaviour) during the first lockdown in Great Britain. First, we aim to assess trends in people’s Capabilities, Opportunities, Motivations, and Behaviours. A decreasing trend in Behaviour could support a descriptive interpretation of behavioural fatigue. Second, we aim to assess the associations between people’s Capabilities, Opportunities, Motivations, and Behaviours.

## 2. Methods

### 2.1. Study Design

The present study is a retrospective analysis of cross-sectional surveys conducted by YouGov. YouGov is an internet-based market research and data analytics company. A search of YouGov’s publications between April 2020 and July 2020 finds eight publications that could have been informed by the present data and could have influenced government policy. None include a comprehensive analysis of behaviour change according to people’s Capabilities, Opportunities, and Motivations. A publication in April 2020 looks at daily moods but not behaviour change [17]. A publication in May 2020 looks at exercise Behaviours but not Capabilities or Opportunities [18]. A publication in June 2020 looks at changes in Behaviour after news leaked of a public official not following the rules, which could change Motivation but not Capability or Opportunity [19]. Additional publications were about charitable donations [20] and stores openings [21,22,23]. The final publication was about whether government policy makes us do what they want us to do, which contained no data analysis [24].

Participation in the present study was voluntary. Participants provided consent for their anonymised data to be used in future analyses and received points they could redeem for prizes on YouGov’s website. The University of Warwick approved this study as a secondary analysis (ID: 110.20-21).

### 2.2. Participants and Setting

The online surveys took place over the first 30 days of the first national lockdown in Great Britain, which started on the 1 April 2020 [25]. YouGov’s recruitment strategy aimed for a representative sample each day in terms of age, gender, social class, and education in terms of the most recent census [26], but these demographics were not connected to individual responses analysed by the present research team. Participants could only take part in one survey. Eligible participants were at least 18 years old and lived in Great Britain. The sample size for the present study was a convenience sample from this retrospective data.

### 2.3. Variables

The selected survey items and their response options appear in Table 1. These survey items were not designed by the present research team but were selected by co-author IV in agreement with the research team to capture each COM-B component. IV has over 25 years of experience conducting behavioural science and policy research. Two items capture the Capability component regarding how knowledgeable participants felt about social distancing and limiting their risks. Two items capture the Opportunity component regarding how easy participants thought it was to follow the social distancing guidelines. Two items capture the Motivation component regarding how worried participants felt about the virus generally and personally. Lastly, four items capture the behaviour change component regarding how often participants were seeing friends, seeing family, going to work, and using public transit. The Opportunity items were only featured in the first 21 days. The remaining items were featured in all 30 days.

### 2.4. Statistical Methods

The items composing each COM-B component were aggregated using means. Participants responding “don’t know” (for the Opportunity or Motivation items) or “not applicable, I did not do this anyway” (for the Behaviour items) were omitted from all analyses. Response scales for Opportunity and Behaviour ranged from 1 to 4, for Motivation ranged from 1 to 5, and for Capability from 1 to 4.5. Higher scores for Capability and Opportunity indicate that participants felt more informed about or had more Opportunities to comply with social distancing guidelines. Higher scores for Motivation indicate that participants felt more concerned about the coronavirus. Higher scores for Behaviour change indicate greater changes in Behaviour compared to before the lockdown commenced.

#### 2.4.1. Tests for Overall Trends

For each COM-B component, graphs displaying the daily mean component scores were visually examined and fitted for best-fit model trend lines. Seven models were compared, including exponential, linear, or polynomial (2nd–6th-degree fits). The model with the largest adjusted R Square and/or the smallest Akaike Information Criterion (AIC)/Bayesian Information Criterion (BIC) scores was selected. When there was a discrepancy between criteria, the BIC was used because it better protects against overfitting [27]. These tests were conducted using Microsoft (R) Excel (R) LTSC MSO (16.0.14332.20771) 64-bit. 

#### 2.4.2. Tests for Relationships

The relationships between the COM-B components were analysed using the Granger’s causality test with tests for stationarity and cointegration using eViews (R) 10+ [28]. Note that assessments for Granger “causality” are not equivalent to randomised controlled trial assessments of causality. Granger’s causality establishes whether the necessary temporal order for causation is present. The test is used for retrospective data analyses. Specifically, it tests whether the preceding levels of two factors, say X and Y, better predict later levels of Y than preceding levels of Y alone. If so, then X is said to “granger-cause” Y [29]. For example, if preceding levels of Capability and Behaviour change are better predictors of Behaviour change than Behaviour change alone, then Capability could be said to “granger-cause” Behaviour change. Granger’s causality test is a particularly powerful associative analysis because the temporal order of granger causes need not be the same in the reverse. That is, while X could granger-cause Y, Y might not granger-cause X.

A flow chart is provided in Figure 2 showing how the data were processed. Each pair of COM-B components were assessed separately. The first series of tests assess whether the data in each pair are stationary, i.e., an assumption for the most straightforward Granger’s causality test. Stationarity is tested for with the Augmented Dickey–Fuller (ADF) with an alpha value of 0.05 (top diamond in Figure 2); the null hypothesis is that the data have a single root, i.e., are not stationary. In many retrospective data sets, particularly in finance and macroeconomics, this assumption is often not met as the data are integrated (Granger, 2003, p. 361) [30]. Here, data were not stationary and, as is common practice, we transformed non-stationary time-series data by first differencing to make the series stationary (Baker, al., 2015, p. 145) [31].

The next series of tests assessed whether the transformed data in each pair were cointegrated using the Johansen Cointegration Test with an alpha value of 0.05 (bottom diamond in Figure 2); the null hypothesis is that there was no cointegration. Where data were not cointegrated, a vector autoregression model was used to assess relationships. Where data were cointegrated, we generated a vector error-correction model. In the vector error correction model, the change in one of the series is explained in terms of the lag difference between the series. If a pair of series is cointegrated at least one lag, one variable is said to be a [granger-]cause of the other (Granger, 2003, p. 366) [30]. In the present analyses, a lag of one day is applied.

## 3. Results

### 3.1. Participants

Of the 49,321 responders who took part in YouGov’s original surveys, 31,359 (63.6%) were omitted from the analyses. Nearly half were omitted (49.0%) for responding “not applicable” to at least one Behaviour item; this response was most common for the Behaviour item about using public transportation (41.1% of all responders), followed by items about going to work (36.2% of all responders), seeing friends (6.7% of all responders) and seeing members of my family they did not live with (6.0% of all responders). Fewer responders were removed for saying “I don’t know” to an Opportunity (5.6% of all responders) or a Motivation (2.2% of all responders) item.

For the remaining 17,962 participants (36.4%), the average number of participants each day was 598.73, with a standard deviation of 28.92.

### 3.2. Outcome of Trend Tests

A summary of the trend analyses for each component of the COM-B model is provided. Tables of the results for the six models compared are in Appendix A.

#### 3.2.1. Capability—Trends Significantly Increasing and Decreasing

The mean daily Capability scores are in Figure 3a. Visual examinations of the graph show participants’ Capabilities scores starting and ending at about 4 out of 4.5. Descriptively, the mean participant had a high sense of knowledge over the first 30 days of lockdown. Statistical tests locate significant increasing and decreasing trends. A third-degree polynomial was the best-fit model (Adjusted R square = 0.39; AIC = −130.64; BIC = −126.43). In the first 7 days, Capability increases. Then, it decreases from days 8 to 24. Lastly, from days 25 to 30, Capability increases again.

#### 3.2.2. Opportunity—Significantly Increasing and Decreasing Trends

The mean daily Opportunity scores are in Figure 3b. Visual examinations of the graph show participants’ Opportunity scores starting and ending at slightly under 3 out of 4. Descriptively, the mean participant thought it was fairly easy to follow the guidelines over the first 30 days of lockdown. Statistical tests locate several significant increasing and decreasing trends. The sixth-degree polynomial was the best-fit model (Adjusted R square = 0.47; AIC = −108.17; BIC = −101.90). Participants reported Opportunity decreases from days 1 to 2. Then, it increases from days 3 to 6 before decreasing from days 7 to 11. Then, Opportunity increases again from days 12 to 16 before again decreasing from days 17 to 20. Lastly, on the 21st day, Opportunity again increases.

#### 3.2.3. Motivation—A Significantly Decreasing Trend

The mean daily Motivation scores are in Figure 3c. Visual examinations of the graph show participants’ Motivation scores starting at around 3.5 out of 5 and slightly decreasing thereafter. The mean participant’s Motivations were between being very and somewhat worried, and slightly decreasing towards being less worried over the first 30 days of lockdown. A linear model was the best-fit model (Adjusted R square = 0.64; AIC = −104.60; BIC = −103.20). The average daily decrease is 0.007.

#### 3.2.4. Behaviour—Trend Significantly Changing

The mean daily Behaviour change scores are shown in Figure 3d. Visual examinations of the graph show participants’ scores starting slightly under 4 out of 4 and slightly decreasing thereafter. The mean participant was becoming slightly less likely to report completely stopping the indicated behaviours over the first 30 days of lockdown. The linear model was the best-fit model (Adjusted R square = 0.17; AIC = −155.79; BIC = −154.39). The average daily decrease was 0.001.

### 3.3. Outcome of Relationship Tests

A summary of the Granger analyses for each pair of COM-B components is provided here. Tables of the results for all tests are provided in Appendix A.

The results of the Augmented Dickey–Fuller test for stationary suggest that Capability (*p* < 0.05) and Behaviour (*p* < 0.05) do not have a unit root; i.e., they are stationary. However, the time series of Opportunity (*p* = 0.12) and Motivation (*p* = 0.43) do. Therefore, following the flow chart in Figure 1, the data were transformed into first differences for the cointegration tests.

The results of the Johansen Cointegration tests suggest that there is no cointegration in four pairs: Capability and Motivation (*p* = 0.62), Capability and Behaviour (*p* = 0.76), Opportunity and Behaviour (*p* = 0.30), and Motivation and Behaviour (*p* = 0.23). For the remaining two pairs, there is cointegration: Capability and Opportunity (*p* < 0.05) and Opportunity and Motivation (*p* < 0.05). Following the above algorithm, the vector autoregression is used for pairs that are not cointegrated. The vector error correction model is used for pairs that are cointegrated.

The relationships revealed in the final analysis are pictured in Figure 4. The direction of the errors indicates the direction of influence. A summary of the effects is provided here: Capability forecasts Opportunity (t = −3.13, *p* < 0.01), but not the reverse; Capability forecasts Behaviour (*t* = −2.41, *p* < 0.05), but not the reverse; and Motivation forecasts Opportunity (*t* = −3.91, *p* = 0.001), but not the reverse. No other relationships were significant.

## 4. Discussion

Across the first lockdown in Great Britain, the current study examines people’s Capabilities, Opportunities, Motivations, and Behaviours. Slight decreases in Behaviour change occurred, which could support a descriptive existence of behavioural fatigue. Granger’s causality tests suggest that Behaviour change could be forecast by slight changes in Capability at the population level. Rather than viewing behavioural fatigue as inevitable, policymakers could have looked at people’s Capabilities to better understand public behaviour and improve public health.

While our trend analyses suggest that slight decreases in Behaviour change occurred, this was not independent of other factors. The present data suggest that Behaviour change could be anticipated by Capability, which is a factor that policy could influence. In this case, communications created about the lockdown rules likely influenced public knowledge that could have influenced adherence (up or down). Other examples exist supporting the notion that policy communications can change behaviour. For example, in August 2020, Parliament announced their “eat out to help out scheme”, which aimed to protect jobs in the hospitality sector by encouraging the public to eat out [32]. Despite little changes in the virus threat from July to August (people’s Motivation to stay at home), more people went out to eat in August and fewer did after the policy stopped in September. Rather than positing an immutable individual construct that influenced eating-out behaviour, the policy enacted and removed provides a more straightforward explanation for the changes in public behaviour.

Conceptualising behavioural fatigue as an immutable factor supports a narrative emphasising individual responsibility and blame. Such a conceptualisation suggests that public health initiatives fail to work because individuals lack sufficient grit to stick out adverse conditions. Grit is thought of as a personality construct with potential genetic dispositions [33]. Belief in grit could foster fatalistic beliefs that there is little policy can do to change people’s behaviour. But this would be a mistake. Even where individual differences exist (even genetic differences), it does not follow that individuals cannot change (at least within some range) [34].

Despite the government and media commonly expressing concerns about people not following lockdown guidance, many people did. King’s College London conducted a cluster analysis that grouped 2250 residents into three segments, including those Accepting, Suffering, and Resisting government actions during the pandemic [35]. Even amongst the Resisting, over 70% reported following government guidance to stay 2 meters away from people outside their home and avoiding places where people gather. One area that received lower adherence across all three segments involved self-isolation. Another study conducted during the first lockdown found that residents with less than 100 GBP in savings were three times less likely to report self-isolating than those with 25,000 GBP or more pounds in savings [36]. Thus, factors beyond individual choice likely contribute to following government guidance. Not considering those factors could decrease the public trust on which public policy must rest [37].

### 4.1. Policy Recommendation

Policymakers could regularly collaborate with data analysts to review and interpret real-time data, ensuring that decisions are informed by quantitative insights and a deep understanding of community contexts. This partnership is crucial, as data do not provide guidance without interpretation. Recommendations regarding how to involve the community in health systems research have been provided elsewhere [38,39]. Where increased adherence coincides with decreases in target public health concerns, such data could also be used to maintain Motivation to follow public health guidance. These Motivational messages could be framed to emphasise that preventive behaviour is within people’s control [40].

Additionally, policymakers could critically assess different conceptual frameworks of behaviour change, integrating contemporary models such as the COM-B model, the Heath Beliefs Model, or the theory of planned behaviour [14,15,16]. Considering each model could facilitate a deeper consideration for the potential consequences of policies aiming to change behaviour in light of economic and social realities. This dynamic approach could better ensure that policymaking is responsive and effective in improving health outcomes.

### 4.2. Strengths and Limitations

The strengths of the current study include the timing of the surveys and the large number of participants. This is a unique data set collected at a pivotal time that could inform policymaking. That said, the data involve participants’ self-reports or perceptions. Other measurements could reveal different relationships. Additionally, the dataset only included the response from the first month of the lockdown period. Further changes may have occurred throughout the pandemic. Demographic information was not available for the research team to analyse and many responders were not included in our analyses. Inferences can be made about those not included. For instance, nearly half of those omitted reported not using public transit before the lockdown. Therefore, we could infer that those removed are more likely to live in areas without reliable public transit (e.g., more rural locations) and are more likely to own a car. Our findings may not generalise to such people.

In our analyses, changes in Behaviour were not predicted by changes in Motivation or Opportunity. Policy actions taken before the lockdown directly impacted Opportunities but did not change greatly over the 30 days these surveys. For example, the Coronavirus Job Retention Scheme was announced on the 20 March 2024, ten days before the lockdown [41]. This scheme provided grants to employers to pay 80% or up to 2500 GBP of furloughed employee’s wages. Employees using that scheme before the lockdown were less likely to experience changes in their Opportunities during the first month of lockdown. Changes in perceived Opportunities were predicted by changes in Motivation, suggesting that Motivation influences how easy people find it to adhere to social distancing rules. More motivated individuals perceive following the rules as easier, which resonates with the evidence [42].

Also not predicted by the original COM-B model, Behaviour did not feedback to predict Capability and Opportunity. As a cross-sectional series of surveys, different individuals completed each survey. While our data show population-level reports of Behaviour change each of the 30 days, we do not have information about individual-level Behaviour change across those 30 days. A repeated measures method could be better suited to assess the feedback loops predicted by the COM-B model. Our retrospective analysis was further limited by the previously collected data, which limits the theories we could consider. We did not have access to data about participant intentions to change their behaviour and so could not evaluate Ajzen’s (1991) [15] theory of planned behaviour. Future studies could include items about participants’ behavioural intentions.

## 5. Conclusions

The current retrospective data analysis examined the relationships between people’s Capabilities, Opportunities, Motivations, and Behaviour change, i.e., adherence to lockdown guidelines across the first 30 days of the lockdown in Great Britain. Though slight, trend changes were identified in all COM-B components. These changes include a decreasing trend in the levels of Behaviour change. Our relationship analyses suggest that the decreases in Behaviour change were not necessarily inevitable. Changes in Capability could be used by policymakers to forecast Behaviour change. Rather than positing immutable factors to explain why residents do (or do not) adhere to government guidance, policymakers could focus on supporting residents’ Capabilities, Opportunities, and Motivations to improve public health.

## Figures and Tables

**Figure 1 behavsci-14-00852-f001:**
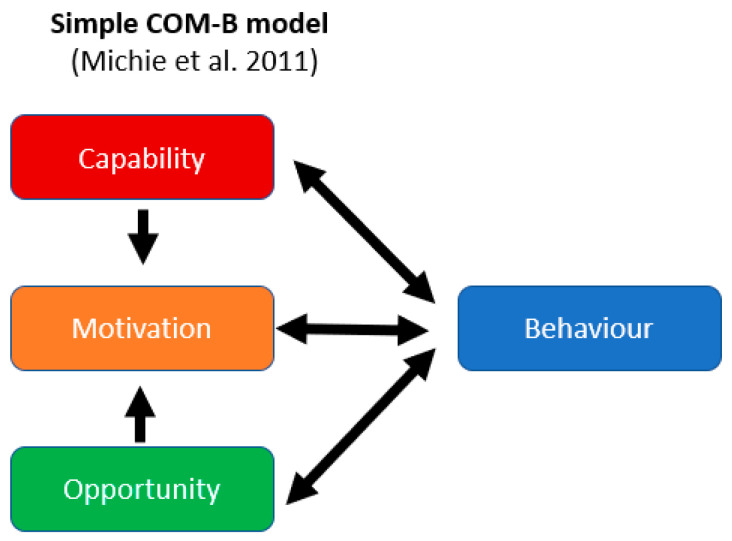
Redrawn COM-B model figures from [12].

**Figure 2 behavsci-14-00852-f002:**
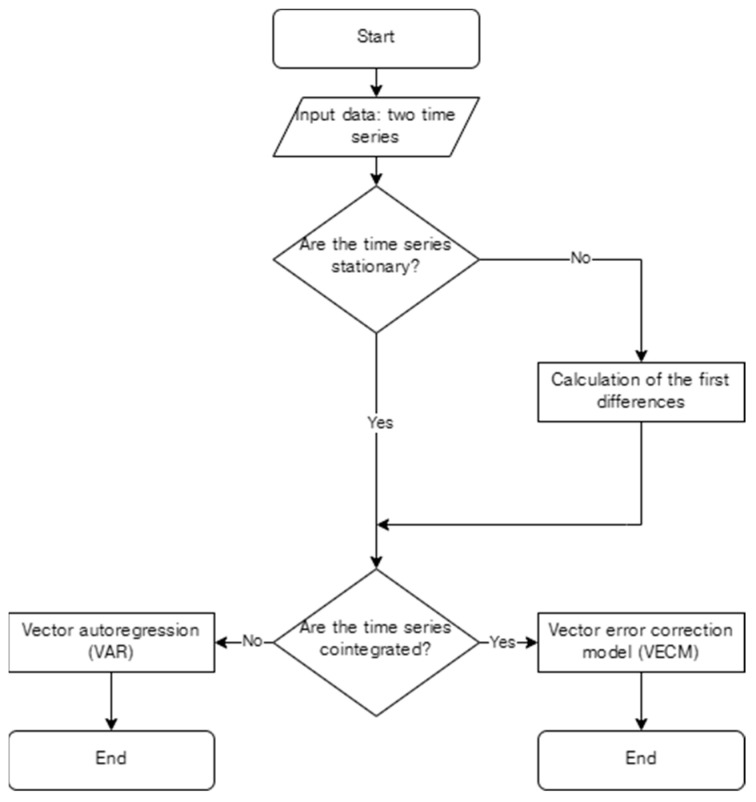
General algorithm of the relationship analyses.

**Figure 3 behavsci-14-00852-f003:**
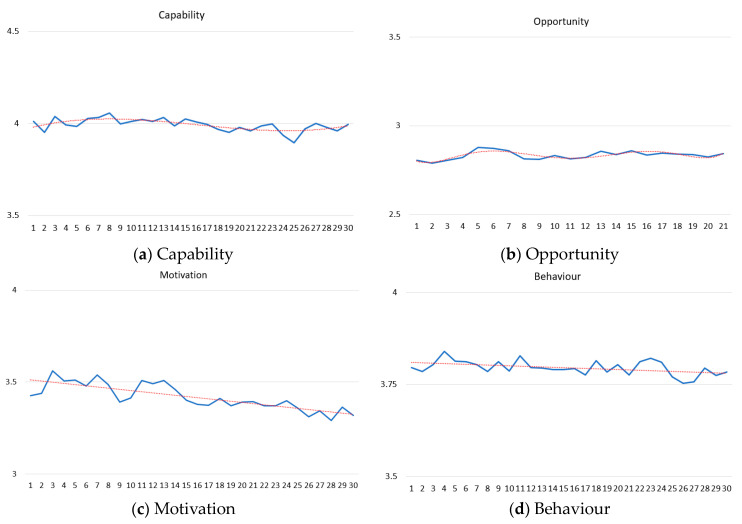
The mean daily scores with a best-fit model line for each COM-B component.

**Figure 4 behavsci-14-00852-f004:**
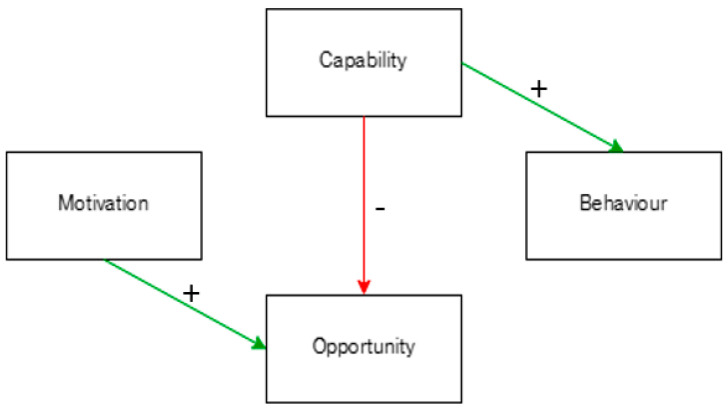
Significant relationships between COM-B components with directionality. Plus signs (green lines) indicate positive relationships, and negative signs (red lines) indicate inverse relationships.

**Table 1 behavsci-14-00852-t001:** Survey Items.

COM-B Component	Item	Response Options as Coded in Present Analyses
Capability	How well informed, if at all, do you feel you are about what social distancing is and how it applies to you?	Very well informed (4), Fairly well informed (3); Not very informed (2); Not informed at all (1)
Capability	I know what I need to do to limit my risk of contracting coronavirus.	Strongly agree (5); Agree (4); Neither agree nor disagree (3); Disagree (2); Strongly disagree (1)
Opportunity	How easy or difficult are you finding it to stick to social distancing rules?	Very easy (4); Fairly easy (3); Fairly difficult (2); Very difficult (1); Do not know (participants removed from analyses)
Opportunity	How easy or difficult do you think other people around you are finding it to stick to social distancing rules?	Very easy (4); Fairly easy (3); Fairly difficult (2); Very difficult (1); Do not know (participants removed from analyses)
Motivation	Overall, how worried are you about coronavirus?	Extremely worried (5); Very worried (4); Somewhat worried (3); Not very worried (2); Not at all worried (1); Don’t know (participants removed from analyses)
Motivation	To what extent do you think coronavirus poses a risk to you personally?	Major risk (5); Significant risk (4); Moderate risk (3); Minor risk (2); No risk at all (1); Don’t know (participants removed from analyses)
Behaviour	Please look at the list of activities below and, for each one, say whether you have reduced how much you are doing it: Seeing Friends	I am still doing this as much as usual (1); I am still doing this, but have cut it down a little (2); I am still doing this, but have cut down a lot (3); I have stopped doing this entirely (4); Not applicable, I did not do this anyway (participants removed from analyses)
Behaviour	Seeing members of my	I am still doing this as much as usual (1); I am still doing this, but have cut it down a little (2); I am still doing this; but have cut down a lot (3); I have stopped doing this entirely (4); Not applicable, I did not do this anyway (participants removed from analyses)
Behaviour	Going to your place of work	I am still doing this as much as usual (1); I am still doing this, but have cut it down a little (2); I am still doing this; but have cut down a lot (3); I have stopped doing this entirely (4); Not applicable, I did not do this anyway (participants removed from analyses)
Behaviour	Using public transit	I am still doing this as much as usual (1); I am still doing this, but have cut it down a little (2); I am still doing this; but have cut down a lot (3); I have stopped doing this entirely (4); Not applicable, I did not do this anyway (participants removed from analyses)

## Data Availability

Data would be made available upon the request with corresponding authors.

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
