# Peer review of "Considering the Role of Behavioural Fatigue in the COVID-19 Lockdown Debates in Great Britain: A Retrospective Analysis of a National Survey Assessing the Relationships between Capability, Opportunity, Motivation, and Behaviour"

_behavsci, 2024, doi:10.3390/bs14090852_

Round 1

Reviewer 1 Report

Comments and Suggestions for Authors

The article is 14 pages long and divided into a few rather typical work parts and subparts. However, the structure and terminology need to be more apparent. A more conventional "Introduction" would make the article more accessible and easier to navigate.

The introduction is an essential part of any research and should answer at least three questions:

·    What the paper is about.

·    Why it is necessary.

·    Why the reader should be interested.

The "Background" starts with the sentence that human behaviours influence the trajectories of infectious disease. This sentence is almost the same as Nita Bharti's in her commentary. This work is the first in the list of references and limits itself to this. In fact, this commentary conveys much more information about the problem and could help introduce it.

The second sentence deals with the statements about changing people's behaviours. Here, the reference point is an excellent paper by Robert West, Susan Mitche, James Rubin, and Richard Amlot. Using interesting papers to back such simple and common-sense statements means losing sight of the broad spectrum of the problem. Instead, the authors move to the following opinion: "Broadly speaking, two policy strategies shaped government interventions, which in line with Rose's seminal work, we/sic!/ referred to as high-risk prevention strategies and population preventive strategies. It looks like the paper by G.Rose from 1985 was the reference point, but that it shaped government intervention is the author's own idea. Such a mixture is very different to accept.

Moreover, the authors are inconsistent because the first sentence of the first paragraph on page 2 clearly states that

Different strategies were implemented across various countries over time.

Such a narrative makes the notion of fatigue unclear, which appears like deus ex machina on the second page. But if one considers the abstract, there is a feeling of mess. There is no 'fatigue' in the title; 'Behaviour fatigue' plays a decisive role in the abstract and again on the second page in light of the COM-B model, a key framework used in this study. The COM-B model, which stands for Capability, Opportunity, Motivation, and Behaviour, needs to be explained or introduced, confusing the reader. A brief explanation of its relevance to the study would help the reader understand its importance.

The management of epidemics or pandemics depends heavily on the adherence of individuals, families, and civil society to the rules and other regulations issued by governments or public health authorities.

Several behaviour models exist, each designed to help us understand what drives behaviour and how and where decisions are made.

The authors describe what they have obtained at the end of this section. Usually, the aims of the planned investigation are included in this section. The authors should reshape this part. Make an authentic introduction to the subject instead of introducing a description of the background. But of what? Such doubt is reasonable because the study's aim is not clearly presented. The authors should consider preparing the typical introduction as the critical factor when discussing editing the article.

Usually, after trying to present the present of work is the second part of the study, namely Materials and Method where

some explanations are needed;

1.    a few words about YouGov and the first investigation,

2.    a few words about select survey items.

The following paragraph is to get acquainted with the Results. No objections; perfectly prepared. The part discussion is not very extensive. Hence, it is not very interesting and results in some exaggeration.

Summarise and propositions.

1.    The management of epidemics and pandemics depends very much on individuals, families, and civil society's adherence to the rules and regulations, generally to the law issued by governments or public health authorities. This adherence can be forced by force through the execution of decrees and other restrictions or may result from different subjects, which ultimately have a decisive role in people's thinking.

2.    A dozen prepared papers confirmed that governments' actions at dawn are imperative and that some acts have limited effect without individual cooperation.

3.    Adherence can be affected by perceived risk in general terms. However, a significant sociocultural factor is social norms and cultural values. Among them is a specific one, namely, "general 'trust in the government and what is of the utmost importance to the prime ministers themselves. Compare the situation, for example, in New Zealand and former Warsaw Pact Countries.

4.    From a more clinical point of view, essential factors are the perceived benefit of some manipulations and the risk of disease outbreaks.

5.    It is very interesting to investigate the linkage between human behaviours and infectious disease, especially when the compliance problem is considered and some people are thinking about other ways to enhance or impede that.

6.    I propose to edit this article with some improvements, the most important of which I was trying to present.

Reviewer 2 Report

Comments and Suggestions for Authors

The paper "Assessing the relationships between capability, opportunity, motivation, and behaviour of people during COVID-19 lockdown in Great Britain: a retrospective analysis of a national survey" has a strong foundation and applies the COM-B mode on large sample which is a strength. The author acknowledged multiple study limitation, which is another strength. I have a few questions for the authors:

You say in the introduction "Compared to other mod els of behaviour (e.g., the common-sense models, the health beliefs model,xiii or the theory of planned behaviour),xiv the COM-B model offers a more comprehensive assessment.xv Capability has to do with an individual’s physical (e.g., skills) or psychological (e.g., awareness) capacity to perform the behaviour."  Could you add a comparative analysis between the COM-B model and other models such as the health belief model and the Theory of Planned Behavior to justify the selection of the COM-B model?

Why did you use a six-degree versus a simpler model? Could you justify? Can you tell the readers why a simpler model was not sufficient? 

In the discussion, could you offer actionable policy recommendations? 
